

# The Doublesex/Mab-3 domain transcription factor DMD-10 regulates ASH-dependent behavioral responses

Julia Durbeck[1], Celine Breton[2], Michael Suter[1], Eric S. Luth[2] and Annette M. McGehee[1]

[1] Biology Department, Suffolk University, Boston, MA, USA
[2] Department of Biology, Simmons University, Boston, MA, USA

## ABSTRACT

The Doublesex/Mab-3 Domain transcription factor DMD-10 is expressed in several cell types in *C. elegans*, including in the nervous system. We sought to investigate whether DMD-10 is required for normal neuronal function using behavioral assays. We found that mutation of *dmd-10* did not broadly affect behavior. *dmd-10* mutants were normal in several behavioral assays including a body bends assay for locomotion, egg laying, chemotaxis and response to gentle touch to the body. *dmd-10* mutants did have defects in nose-touch responsiveness, which requires the glutamate receptor GLR-1. However, using quantitative fluorescence microscopy to measure levels of a GLR-1::GFP fusion protein in the ventral nerve cord, we found no evidence supporting a difference in the number of GLR-1 synapses or in the amount of GLR-1 present in *dmd-10* mutants. *dmd-10* mutants did have decreased responsiveness to high osmolarity, which, along with nose-touch, is sensed by the polymodal sensory neuron ASH. Furthermore, mutation of *dmd-10* impaired behavioral response to optogenetic activation of ASH, suggesting that *dmd-10* promotes neuronal signaling in ASH downstream of sensory receptor activation. Together our results suggest that DMD-10 is important in regulating the frequency of multiple ASH-dependent behavioral responses.

Corresponding author
Annette M. McGehee,
amcgehee@suffolk.edu

## INTRODUCTION

DMD-10 is a member of the Doublesex/Mab-3 domain family of transcriptional regulators. Proteins in this family exhibit sequence specific DNA binding activity through a zinc-binding DM domain (*Zhu et al., 2000*). Much of the research on DM domain proteins relates to their role in sex determination and sexual differentiation (*Matson & Zarkower, 2012*; *Zarkower, 2001*); however, DM domain family members can be expressed in tissues other than the gonads (*Hong, Park & Saint-Jeannet, 2007*). DM domain proteins have been found to be important in the nervous system; both in regulating neuronal development (*Saulnier et al., 2013*; *Tresser et al., 2010*; *Yoshizawa et al., 2011*), and in regulating behavior (*Andersson et al., 2012*).

In *C. elegans* there are 11 members of the DM domain (DMD) family. While some DMD family members have been extensively studied (*Matson & Zarkower, 2012*), to date

nothing has been published about the function of DMD-10. Analysis of tissue specific expression studies (*Kaletsky et al., 2018*; *Spencer et al., 2011*) indicates that *dmd-10* is likely to be expressed in several different cell types across *C. elegans* including the intestine, muscles, coelomocytes and neurons. These expression studies suggest that *dmd-10* may be expressed in a number of different neuron types in hermaphrodites, which raises the question of whether *dmd-10* is required for normal neuronal functioning.

Previous studies in *C. elegans* support the idea that DMD family members can have important roles in controlling neuronal function. DMD-3 is required in the PHC neuron for correct axonal extension and upregulation of presynaptic machinery in males (*Serrano-Saiz et al., 2017*). DMD-5 and DMD-11 are required for the establishment of the appropriate synaptic connectivity of neurons PHB and AVG in males (*Oren-Suissa, Bayer & Hobert, 2016*). While roles for some DMD family members have been identified, DMD-10 is currently uncharacterized.

We sought to determine whether DMD-10 is required for normal neuronal function in hermaphrodites, the predominant naturally-occurring sex. One of the benefits of using *C. elegans* to identify and characterize regulators of neuronal signaling is that there are simple, quantifiable behaviors associated with the activity of defined neuronal circuits (*Hart, 2006*). Behaviors such as egg-laying, chemotaxis, and body bending are associated with signaling in particular circuits (*Hart, 2006*; *Hobert, 2003*). By assessing the range of behaviors that are affected in a particular mutant strain it is possible to determine the neuronal circuits in which that gene normally functions. For example, mutants that are defective in chemotaxis towards specific chemicals indicate a defect in the chemosensory circuit. Mutants of *odr-10* are deficient in chemotaxis towards diacetyl, a chemical sensed by the AWA neuron, suggesting a role for *odr-10* in this chemosensory circuit (*Sengupta, Chou & Bargmann, 1996*). Indeed, *odr-10* encodes a receptor that is expressed in AWA and is required for sensing the presence of diacetyl (*Sengupta, Chou & Bargmann, 1996*). In a different neuronal circuit, *glr-1* mutants are defective in their response to a touch delivered to the nose tip (*Hart, Sims & Kaplan, 1995*; *Maricq et al., 1995*). In the case of GLR-1 this behavioral defect is not due to a defect in the sensory neurons of the circuit, but rather to a defect in the command interneurons where the GLR-1 glutamate receptor is required in order to signal for reversals in response to nose touch stimuli (*Hart, Sims & Kaplan, 1995*; *Maricq et al., 1995*). We employed these and several other behavioral assays in order to determine where in the nervous system DMD-10 may function.

In this study we asked whether DMD-10 plays an important role in the nervous system by performing a behavioral survey of *dmd-10* mutants. We found that *dmd-10* is not broadly required for neuronal function, as many behaviors are normal in these mutants. We did find that *dmd-10* mutants have a defect in some behaviors that are mediated by ASH sensory neurons. ASH neurons are a pair of polymodal sensory neurons that mediate *C. elegans'* response to and avoidance of several nociceptive stimuli including touch to the nose, high osmolarity, pH and other aversive chemicals through specialized dendritic sensory cilia (*Bargmann, Thomas & Horvitz, 1990*; *Hilliard, Bargmann & Bazzicalupo, 2002*; *Hilliard et al., 2004*; *Kaplan & Horvitz, 1993*; *Troemel et al., 1995*).
Additional assays suggest that DMD-10 may regulate these behaviors by influencing presynaptic signaling from ASH sensory neurons.

## MATERIALS AND METHODS

### Strains and strain maintenance

*C. elegans* were raised under standard laboratory conditions at 20 °C on nematode growth media (NGM) agar plates with OP50, a strain of *Escherichia coli* as previously described (*Brenner, 1974*). Strains were not backcrossed prior to use. The *C. elegans* strains used in this study and from where they were obtained are as follows: CGC: N2 (wild type strain), *glr-1(n2461)*, *dmd-10(gk1131)*, *dmd-10(gk1132)*, *dmd-10(gk1125)*, *egl-10(n692)*, *mec-10 (e1515)*, *osm-10(n1602)*, *tph-1(mg280)*, *unc-51(e369)*, *odr-10(ky225)*, William Schafer: *ljIs114 (gpa-13p::*FLPase, *sra-6p::FTP::ChR2::YFP)*, Peter Juo: *ljIs114; eat-4*, Joshua Kaplan: *nuIs24 (glr-1p::GLR-1::GFP)*, *nuIs80 (glr-1p::GLR-1(A/T)::YFP)*, This study: *nuIs24; dmd-10(gk1131)*, *nuIs80 dmd-10(gk1131)*, *ljIs114; dmd-10 (gk1131)*.

### Behavioral assays

All behavioral assays were performed blinded to the genotype of the animals.

#### Egg laying assays

Egg laying assays on N2, *dmd-10(gk1131)* and *tph-1(mg280)* animals were performed as described (*Carnell et al., 2005*). Aged matched adults were prepared by plating L4s onto OP50-seeded NGM plates 24 h prior to the assay. To later corral the animals on the NGM assay plate, one 16 mm copper ring (PlumbMaster catalog # 17668) for each genotype was dipped in ethanol, heated for 5 s using a Bunsen burner flame, and placed on the surface of the same assay 60 mm plate such that it became lightly embedded on the plate surface. A total of 20 microliters of an overnight OP50 culture was spotted in the center of each ring and allowed to dry under a flame for 10–30 min before 10 adults were added to the dry OP50 spot. The number of eggs laid by each group was counted after 60 min and the number of eggs laid per individual was calculated. In this manner, groups of all three genotypes were assayed on the same plate at the same time for each experiment.

#### Thrashing

Thrashing assays were performed on young adult N2, *dmd-10(gk1131)*, and *egl-10(n692)* animals as previously described (*Hart, 2006*; *Miller et al., 1996*). Briefly, the thrashing of individuals in M9 buffer was assayed using the wells of a 96-well plate. Individuals were allowed to acclimate for 20 s, then were observed for 1 min. Body bends (thrashes) were counted each time the body was maximally curved in a particular direction. The number of body bends for each individual was recorded.

#### Nose touch

Nose touch behavioral assays were performed on young adult N2, *glr-1(n2461)*, *dmd-10 (gk1131)*, *dmd-10(gk1132)* and *dmd-10(gk1125)* animals as previously described (*Kaplan & Horvitz, 1993*). Assay plates were seeded with a 1:10 dilution of an overnight OP50 culture in Lennox broth (LB) 1 day before the assay. One individual was transferred to the

center of the thinly-seeded assay plate. An eyelash was placed a short distance in front of a forward-moving animal. It was noted whether the animal reversed its locomotion (response) or did not (non-response) immediately upon contacting the eyelash at a 90° angle. A reversal was defined as backward movement greater than the distance from the nose to the terminal bulb of the pharynx. Each individual was tested with 10 eyelash encounters, the timing between repeat encounters varied, usually between 10 and 20 s. The response for each touch for each individual was recorded. As there were no differences in the responses across the ten encounters average responses across the 10 trials are reported.

### Gentle touch

Gentle touch assays were performed on young adult N2, *dmd-10(gk1131)*, and *mec-10 (e1515)* animals as previously described (*Hobert et al., 1999*). To begin the assay, one individual was placed on an NGM agar plate with OP50. An eyelash was used to alternately stroke the anterior and posterior body of the animal. A total of 10 touches were made (five anterior and five posterior), and changes in locomotor direction were scored as a response. The alternating touches were delivered in rapid succession with 1–2 s between each touch. The response for each trial for each individual was recorded. Responses did not vary across the series of touches, average responses are reported.

### Dry drop

Dry drop behavioral assays were performed on young adult N2, *glr-1(n2461)*, *dmd-10 (gk1131)*, and *osm-10(n1602)* animals with some variations from a previously described assay (*Hilliard, Bargmann & Bazzicalupo, 2002*). Before testing, one drop of an 8M glycerol with 0.02% bromophenol blue was spotted onto an unseeded NGM assay plate. Once the drop was dry, one individual was added to the assay plate near the drop. Data was recorded for whether or not the animal reversed after encountering the glycerol/dye mixture. Ten encounters for each individual were observed, the timing between encounters varied as animals were allowed to roam freely on the plate and encounter the drop in the course of their natural movement. The response for each encounter for each individual was recorded. As responses did not vary across the series of encounters average responses are reported.

### Optogenetic activation of ASH

Optogenetic activation of ASH (OptoASH) was performed similarly to what has been previously described (*Schmitt et al., 2012*; *Luth et al., 2021*). *ljIs114, ljIs114; eat-4 (n2474), and ljIs114; dmd-10(gk1131)* animals used for OptoASH assays were grown for at least one generation in the dark on NGM agar plates seeded with OP50 mixed with 100 μM all-trans-retinal (ATR). For behavioral testing young adults were transferred to an NGM agar plate spotted with OP50 but lacking ATR. Transfer to plates without ATR immediately before testing still yields robust reversal responses in WT animals and reveals expected reversal phenotypes for a variety of mutants with known defects in the classical, eyelash-stimulated nose touch response (*Luth et al., 2021*). Animals were illuminated from a distance of approximately 10 cm using a dynamic LED light kit containing a 1-up CREE

XT-E royal blue LED equipped with a Carclo 20 mm narrow spot LED optic (LED Supply). The measured light intensity at the plate was 0.47 mW/mm$^2$. Reversals were triggered using 1-s pulses of blue light with an interstimulus interval of 10 s (*Schmitt et al., 2012*). Locomotor reversals (backward movement greater than the distance from the nose to the terminal bulb of the pharynx) observed during or immediately after illumination was counted as a response. The number of reversals per 10 simulations for each individual was recorded.

### Chemotaxis

Chemotaxis assays were performed on N2, *odr-10 (e1515)*, and *dmd-10 (gk1131)* animals similarly to previously described (*Bargmann, Hartwieg & Horvitz, 1993*). Assay plates were prepared 2 days prior to the assay 10 ml of CTX agar (2% agar, 5 mM KPO$_4$ pH = 6, 1 mM CaCl$_2$, 1 mM MgSO$_4$) in 10 cm plates. The day of the assay CTX plates were spotted with 1 μl of 1 M NaN$_3$ on opposite sides of the plate. The same locations were then spotted with 1 μl of either chemoattractant (0.1% diacetyl or 1% isoamyl alcohol (Sigma-Aldrich, St. Louis, MO, USA)) or vehicle (ethanol). Assays were performed using populations of *C. elegans* that had been laid 3 days prior to the experiment. The *C. elegans* were washed twice with M9 buffer and once with distilled deionized water. Between 50 and 300 individuals were pipetted on the midline of the plate and excess liquid was removed with a kimwipe. After 1-h incubation at 20 °C the number of individuals within a 2 cm circle centered on each spot (chemoattractant and vehicle) and the total number of individuals on the plate were recorded (individuals that had not moved after being placed on the plate were not counted).

### Spontaneous reversals

Spontaneous reversal assays were performed on young adult *nuIs80* and *nuIs80 dmd-10 (gk1131)* animals as previously described (*Juo et al., 2007*; *McGehee, Moss & Juo, 2015*). Individuals were transferred to unseeded NGM assay plates using halocarbon oil and allowed to acclimate for 2 min. The number of reversals (transitions from forward to backward movement) made over 5 min was recorded, and the number of reversals per minute was calculated.

## Fluorescence imaging and analysis

Fluorescence imaging was performed as previously described (*Kowalski, Dahlberg & Juo, 2011*). Briefly, *nuIs24* or *nuIs24; dmd-10(gk1131)* young adults were immobilized using 30 mg/ml 2,3-butanedione monoxamine (Sigma-Aldrich, St. Louis, MO, USA) and the anterior ventral nerve cord immediately posterior to RIG was imaged (*Juo et al., 2007*). A total of 1 μm (total depth) Z-series stacks were collected using a Carl Zeiss Axiovert M1 microscope with a 100X Plan Apochromat (1.4 numerical aperture) objective equipped a GFP filter cube. Images were collected with an Orca-ER CCD camera (Hamamatsu) and MetaMorph (version 7.1) software (Molecular Devices). Maximum intensity projections of Z-series stacks were used for quantitative analyses of fluorescent puncta. Exposure settings and gain were adjusted to fill the 12-bit dynamic range without saturation and were identical for all images taken. Line scans of ventral cord puncta were

generated using Meta-Morph (version 6.0) and were analyzed using custom written software (*Burbea et al., 2002*) with Igor Pro (version 6 Wavemetrics). Statistical analyses were performed using Student's *t* test.

## Statistics and analysis

The results of behavioral assays were graphed and statistically analyzed using Prism version 8 (GraphPad). Data was analyzed using a one-way ANOVA followed by a Tukey's multiple comparison test comparing different genotypes, except where indicated.

## RESULTS

We are interested in understanding the role of the Doublesex/Mab domain transcription factor DMD-10 in neuronal function as it relates to behavior. *dmd-10* mRNA has been found to be enriched in neurons (*Kaletsky et al., 2018*; *Spencer et al., 2011*) indicating a potential neuronal function. We tested for possible roles of DMD-10 in normal neuronal function using different behavioral assays. In order to do this we used the *dmd-10(gk1131)* deletion allele that removes the first three exons of the gene, including approximately half of the putative DNA binding domain, and is likely to be a functional null (Fig. 1A).

We first tested motor neuron functionality. Motor neurons stimulate muscle contraction by releasing the neurotransmitter acetylcholine, and genes that affect this neuromuscular signaling have changes in the frequency of rhythmic movements of the body, called thrashing (*Miller et al., 1996*). One such gene is *egl-10*, a regulator of G protein signaling (*Koelle & Horvitz, 1996*). *egl-10* mutants show decreased rates of locomotion and thrashing (*Koelle & Horvitz, 1996*; *Miller et al., 1996*). Thrashing behavior can be measured by placing an individual in a drop of buffer and recording the number of body bends (*Miller et al., 1996*). The number of bends per minute was counted for wild type, *dmd-10*, and *egl-10* mutants (Fig. 1B). As expected, we found a significant decrease in the number of body bends per minute for *egl-10* mutant as compared to the wild-type controls (Fig. 1B). However, we saw no difference in the number of body bends per minute in *dmd-10* mutants as compared to the wild-type controls (Fig. 1B). This suggests that there is not a gross locomotor defect in these mutants.

Along with unaltered behavior driven by motor neurons controlling locomotion, we also found no defect in the motor control of egg laying. Egg laying behavior is principally mediated by the serotonergic HSN motor neurons. Serotonin release by HSN neurons directly modulates $Ca^{2+}$ transients in vulval muscles to alter the rate of egg laying (*Shyn, Kerr & Schafer, 2003*; *Waggoner et al., 1998*). We observed no difference in the rate of egg laying in *dmd-10* adults compared to wild type adults, while, as expected, egg laying was significantly reduced in *tph-1* mutants deficient in serotonin biosynthesis (Fig. 1C) (*Carnell et al., 2005*).

We next wanted to test the sensory responses of *dmd-10* mutants. *C. elegans* are able to sense and respond to volatile organic compounds using the sensory neurons AWA and AWC (*Bargmann, 2006*). The ability to respond to olfactory cues can be measured using chemotaxis assays in which movement of animals towards or away from a point source of a chemoattractant is measured. Animals with mutations that alter the functioning of

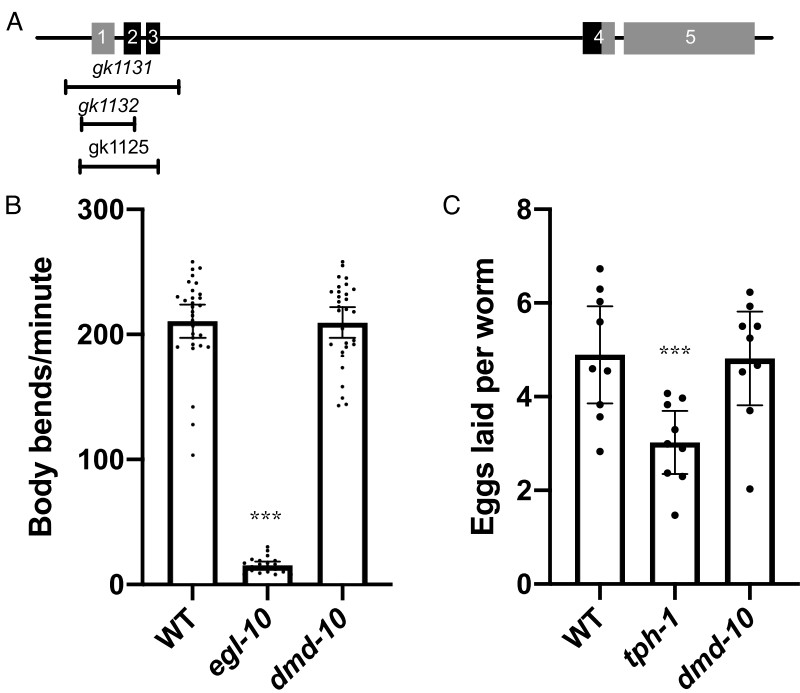

**Figure 1** ***dmd-10* mutants have normal motor behaviors.** (A) Schematic of the *dmd-10* gene locus showing the locations of the five exons and the three deletion alleles that were tested. Boxes 1–5 indicate the 5 exons of the gene. The portions of the exons shown in black indicate the location of the DNA binding domain. (B) Graph showing the average number of body bends per minute for the following genotypes: WT $n = 30$, *egl-10(n692)* $n = 20$ and *dmd-10(gk1131)* $n = 30$ (Tukey's multiple comparisons test: WT vs *egl-10* $p < 0.0001$, WT vs *dmd-10* $p = 0.989$, *egl-10* vs *dmd-10* $p < 0.0001$). Each dot represents a single animal. Error bars denote 95% confidence intervals. (C) Graph showing the mean number of eggs laid per animal in 60 min over nine independent experiments for WT, *tph-1(mg280)* and *dmd-10(gk1131)* (Tukey's multiple comparisons test: WT vs *tph-1* $p = 0.0075$, WT vs *dmd-10* $p = 0.989$, *tph-1* vs *dmd-10* $p = 0.0105$). Each dot represents the mean of 10 animals. Error bars denote 95% confidence intervals. ***$p < 0.0001$.

these sensory circuits have defects in chemotaxis. For example, ODR-10 is a G-protein coupled receptor expressed in AWA sensory neurons that binds diacetyl and is required for its detection, *odr-10* mutants are defective in chemotaxis towards diacetyl (*Sengupta, Chou & Bargmann, 1996*). ODR-10 is not required for the detection of isoamyl alcohol which is instead sensed by the AWC sensory neurons, and thus *odr-10* mutants are not defective in chemotaxis towards isoamyl alcohol (*Bargmann, Hartwieg & Horvitz, 1993*; *Sengupta, Chou & Bargmann, 1996*). To assess the ability of *dmd-10* mutants to respond to olfactory cues, we tested their response to the chemoattractants diacetyl and isoamyl alcohol (Figs. 2A and 2B). As expected *odr-10* mutants showed defective chemotaxis towards diacetyl (Fig. 2A), but no defect in chemotaxis towards isoamyl alcohol (Fig. 2B). The *dmd-10* mutants had chemotactic responses that were indistinguishable from wild type (Figs. 2A and 2B). These results suggest that *dmd-10* is not required for chemotaxis toward diacetyl or isoamyl alcohol.

We next wanted to test mechanosensory responses. In *C. elegans* gentle touch to the body is sensed through an anterior sensory circuit where touch is sensed by the ALM and AVM sensory neurons, and a posterior sensory circuit where touch is sensed by the PLM

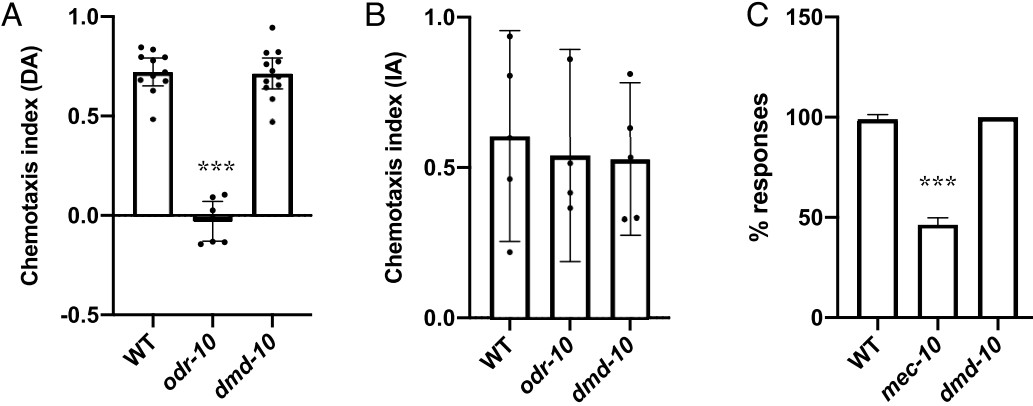

**Figure 2 Sensory responses are normal in *dmd-10* mutants.** (A and B) Chemotaxis assays were performed on populations of WT, *odr-10(ky225)* and *dmd-10(gk1131)*. The chemotaxis index (CI) for each plate was calculated. CI = (# at test − # at control)/total #. Each dot represents an independent assay plate. Error bars denote 95% confidence intervals. (A) Chemotaxis towards diacetyl WT *n* = 11, *odr-10 n* = 7, *dmd-10 n* = 12 (Tukey's multiple comparisons test WT vs *odr-10 p* < 0.0001, WT vs *dmd-10 p* = 0.988, *odr-10* vs *dmd-10 p* < 0.0001). (B) Chemotaxis towards isoamyl alcohol WT *n* = 5, *odr-10 n* = 4, *dmd-10 n* = 5 (Tukey's multiple comparisons test WT vs *odr-10 p* = 0.918, WT vs *dmd-10 p* = 0.871, *odr-10* vs *dmd-10* p=0.997). (C) Individuals were assayed using 10 sequential touches to the body, alternating between anterior and posterior touches. Graphs show the average percentage of responses. Error bars denote 95% confidence intervals. (WT *n* = 10, *mec-10(e1515) n* = 11, *dmd-10(gk1131) n* = 9. Tukey's multiple comparisons test WT vs *mec-10 p* < 0.0001, WT vs *dmd-10 p* = 0.816, *mec-10* vs *dmd-10 p* < 0.0001). ***$p$ < 0.0001.           

sensory neuron (*Chalfie & Sulston, 1981*; *Chen, Cuadros & Chalfie, 2015*). This touch is sensed in part through mechanoreceptors that include the mechanoreceptor subunit MEC-10, and *mec-10* mutants have decreased sensitivity to touch (*Árnadóttir et al., 2011*; *Chalfie & Sulston, 1981*). We used a gentle touch assay in order to test mechanosensory responses (*Hobert et al., 1999*). In this assay individuals are repeatedly touched, alternating between touches to the anterior and posterior body. Wild-type animals respond with a change in locomotion direction (for example, anterior touch leads to backward movement). As expected, mutation in the mechanoreceptor subunit *mec-10* decreased sensitivity to touch (*Árnadóttir et al., 2011*; *Chalfie & Sulston, 1981*), while the *dmd-10* mutants had overall touch response rates that were indistinguishable from wild type (Fig. 2C). These results suggest that DMD-10 is not required to respond to gentle touch to the body.

We next performed a nose touch behavioral assay. Gentle touch to the nose activates a glutamatergic mechanosensory reflex leading to a reversal of locomotion from forward to backward (*Kaplan & Horvitz, 1993*; *Lee et al., 1999*). Response to nose touch requires the glutamate receptor GLR-1 and is therefore significantly decreased in *glr-1* mutants compared to wild-type (*Hart, Sims & Kaplan, 1995*; *Kowalski, Dahlberg & Juo, 2011*; *Maricq et al., 1995*). Indeed, in our assay we detected no responses to nose touch in *glr-1* mutants (Fig. 3). Nose touch responsiveness was also significantly reduced in *dmd-10 (gk1131)* mutants as compared to wild type (Fig. 3).

To confirm that the defect in nose touch responsiveness in the *dmd-10 (gk1131)* mutants was due to the mutation in *dmd-10* and not to a background mutation, we tested

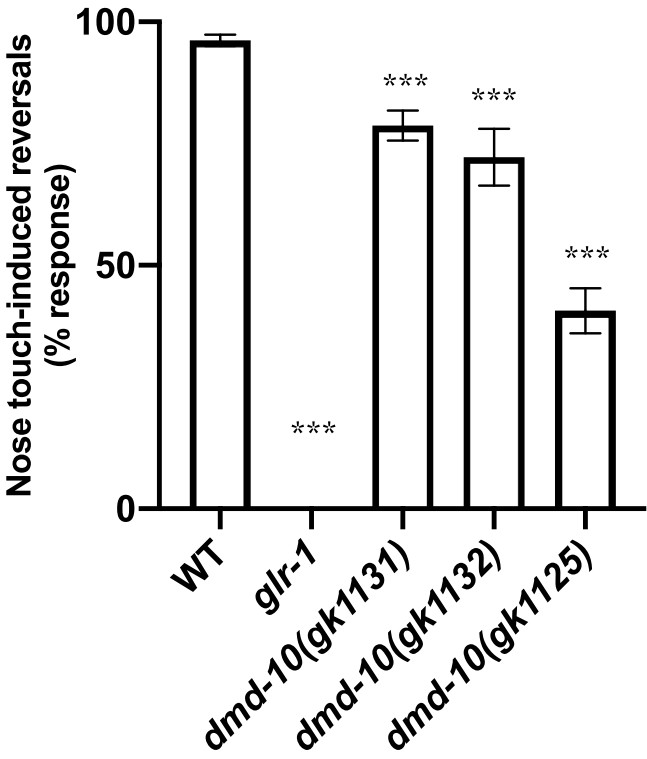

**Figure 3** ***dmd-10*** **mutants have a decreased response to nose touch.** Reversal responses of individuals to nose touch trials. Graph showing percentage of responses out of 10 trials per individual for the following genotypes: WT *n* = 126, *glr-1(n2461)* *n* = 104, *dmd-10(gk1131)* *n* = 142, *dmd-10(gk1132)* *n* = 52, and *dmd-10(gk11125)* *n* = 45. Tukey's multiple comparisons test: WT vs *glr-1* *p* < 0.0001, WT vs *dmd-10* (gk1131) *p* < 0.0001, WT vs *dmd-10* (gk1132) *p* < 0.0001, WT vs *dmd-10(gk1125)* *p* < 0.0001, *glr-1* vs *dmd-10(gk1131)* *p* < 0.0001, *glr-1* vs *dmd-10(gk1132)* *p* < 0.0001, *glr-1* vs *dmd-10(gk1125)* *p* < 0.0001, *dmd-10(gk1131)* vs *dmd-10(gk1132)* *p* = 0.029, *dmd-10(gk1131)* vs *dmd-10(gk1125)* *p* < 0.0001, *dmd-10* (gk1132) vs *dmd-10(gk1125)* *p* < 0.0001. Error bars denote 95% confidence intervals. ***$p$ < 0.0001.

additional *dmd-10* mutant alleles *(gk1132* and *gk1125)*. Both of the tested alleles also responded to nose touch less frequently than wild-type (Fig. 3). While the *dmd-10(gk1125)* allele showed the greatest difference from the wild type response rate, these worms exhibited additional phenotypes including slow growth and sluggish locomotion that were not present in the two other *dmd-10* mutant strains. These additional phenotypes and the fact that the nose-touch phenotype of the *dmd-10(gk1125)* strain is significantly decreased from both other *dmd-10* alleles lead us to speculate that the *dmd-10(gk1125)* strain has an additional mutation(s) at a genetic locus other than the *dmd-10* locus that accounts for the stronger nose-touch response defect we saw in this strain. Since each of the tested *dmd-10* alleles independently showed a decreased nose touch responsiveness it is likely that this phenotype is in fact due to the lack of functional DMD-10 in these strains.

One possible explanation for a reduced nose touch response rate is a reduction in GLR-1 receptor levels (*Juo et al., 2007*; *Kowalski, Dahlberg & Juo, 2011*). In order to directly compare GLR-1 levels in wild-type and *dmd-10* mutants we measured the levels of GFP-tagged GLR-1 (GLR-1::GFP) in the anterior ventral nerve cord immediately posterior
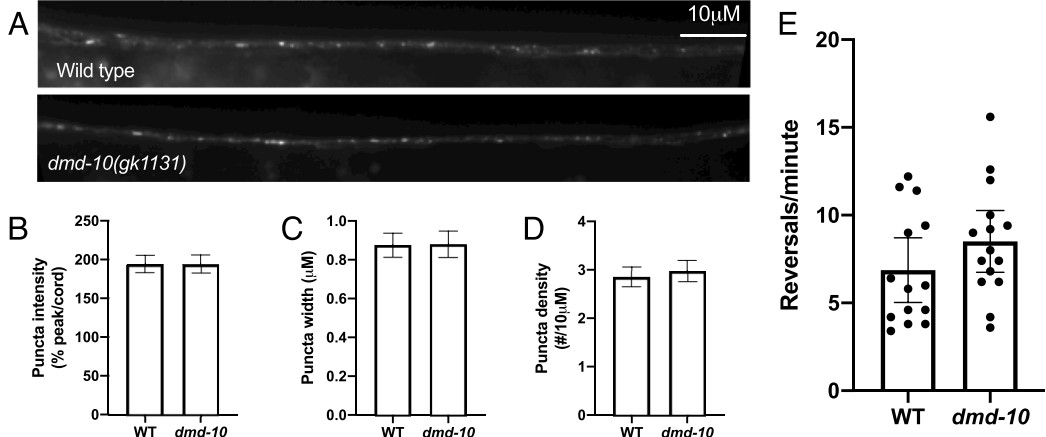

**Figure 4 GLR-1::GFP levels are normal in *dmd-10* mutants.** (A) Representative images of GLR-1::GFP in the ventral nerve cord of WT and *dmd-10(gk1131)* worms. (B–D) Quantitative analysis of WT (*n* = 23) and *dmd-10* (*n* = 24) images. Error bars denote SEM. (B) Quantitative analysis of puncta intensity (relative to background fluorescence in the ventral nerve cord) (*p* = 0.998), (C) puncta width (*p* = 0.918), and (D) puncta density (*p* = 0.405). (E) Spontaneous locomotor reversals of WT (*n* = 14) and *dmd-10* (*gk1131*) mutants (*n* = 15) expressing GLR-1(A/T) (*p* = 0.179). Student's *t* test. Each dot represents a single animal. Error bars denote 95% confidence intervals.

to the RIG neuron. The GLR-1::GFP fusion protein is functional and localizes to postsynaptic sites along interneurons of the *C. elegans* ventral nerve cord (*Burbea et al., 2002*; *Rongo et al., 1998*) (Fig. 4A). In other *C. elegans* mutant strains that have decreased responses to nose touch, lower levels of GLR-1::GFP are detected in the anterior ventral nerve cord (*Juo et al., 2007*; *Kowalski, Dahlberg & Juo, 2011*). We found no significant differences in GLR-1::GFP puncta density, intensity, or width in *dmd-10* mutants as compared to wild-type (Figs. 4B–4D). These results suggest that there are no differences in the number of GLR-1-containing synapses or in the amount of GLR-1 that is localized to ventral nerve cord synapses in the *dmd-10* mutants.

While we saw no change in GLR-1::GFP levels, it is possible that the GLR-1::GFP we detected in the ventral nerve cord is not all localized to the plasma membrane or at synapses. Mutations that alter the levels of GLR-1 at the cell surface show altered behavioral phenotypes, even when total levels of GLR-1 are not decreased (*Juo & Kaplan, 2004*; *Rongo et al., 1998*; *Schaefer & Rongo, 2006*; *Zhang et al., 2012*). Therefore, the nose touch defect in *dmd-10* mutants could be the result of inappropriately localize GLR-1 that cannot be activated by presynaptic glutamate release. We tested this possibility using an additional behavioral assay that requires functional and properly-localized GLR-1, namely reversals during spontaneous locomotion. Spontaneous reversal frequency depends on the strength of glutamate signaling, and alterations in GLR-1 function produce corresponding changes in reversal rate (*Zheng et al., 1999*, *2004*). In order to look at GLR-1 function independent of presynaptic glutamate release, we took advantage of a constitutively active version of the GLR-1 protein (GLR-1(A/T)) (*Zheng et al., 1999*). Individuals expressing GLR-1(A/T) reverse frequently even in the absence of glutamate release (*Zheng et al., 1999*), while mutations that affect GLR-1 function can suppress these

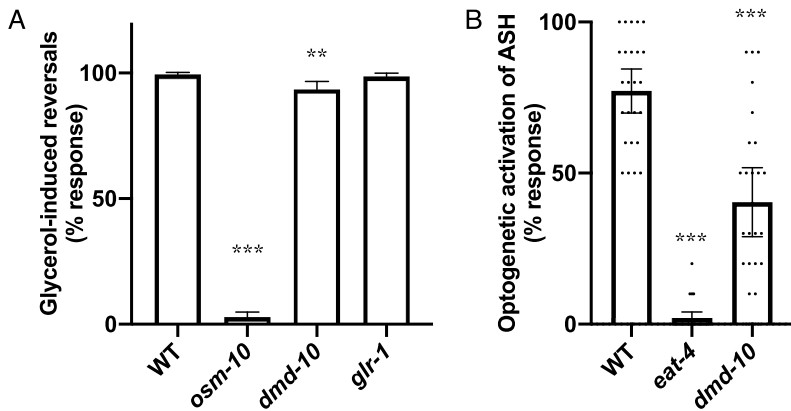

**Figure 5 Decreased ASH responses in *dmd-10* mutants.** (A) Reversal responses of *C. elegans* encountering a dried drop of 8M glycerol. WT *n* = 39, *osm-10(n1602) n* = 35, *dmd-10(gk1131) n* = 35, and *glr-1(n2461) n* = 30. Error bars denote 95% confidence intervals. Tukey's multiple comparisons test: WT vs *osm-10 p* < 0.0001, WT vs *dmd-10 p* = 0.0001, WT vs *glr-1 p* = 0.939, *osm-10* vs *dmd-10 p* < 0.0001, *osm-10* vs *glr-1 p* < 0.0001, *dmd-10* vs *glr-1 p* = 0.0026. (B) Reversal responses of *C. elegans* expressing channelrhodopsin 2 in ASH to pulses of 450 nm light. WT *n* = 25, *eat-4 n* = 25, and *dmd-10(gk1131) n* = 26. Tukey's multiple comparisons test: WT vs *eat-4 p* < 0.0001, WT vs *dmd-10 p* < 0.0001, *eat-4* vs *dmd-10 p* < 0.0001. Each dot represents a single animal. Error bars denote 95% confidence intervals. \*\**p* < 0.001, \*\*\**p* < 0.0001.

reversals (*Zheng et al., 2004*). If the *dmd-10* mutation causes GLR-1 to be inappropriately retained in internal compartments within neurons, then we would expect that the GLR-1 (A/T) protein would be similarly mislocalized and unable to signal. Therefore, there would be a corresponding decrease in the spontaneous reversal rate in the *dmd-10* mutants. We did not see a decrease in the reversal rate of *dmd-10* mutants expressing GLR-1(A/T) compared to wild-type animals expressing GLR-1(A/T) (Fig. 4E), suggesting that in the *dmd-10* mutants the GLR-1(A/T) protein is at the cell surface and is fully functional. Together, the imaging and reversals data suggest that the reduction in nose touch response rate that we observed in the *dmd-10* mutants is not due to a defect in GLR-1 regulation.

Since we did not find any evidence that the ASH-mediated nose-touch defect in *dmd-10* mutants is caused by a defect in GLR-1 in downstream neurons we next looked at additional ASH-mediated behavior. ASH neurons are polymodal sensory neurons that sense several different aversive stimuli including high osmolarity (*Kaplan & Horvitz, 1993*). Sensation of high osmolarity is likely transduced by the OSM-10 accessory subunit of the OSM-9/OCR-2 receptor expressed on ASH cilia, and *osm-10* mutants are defective in responding to high osmolarity (*Colbert, Smith & Bargmann, 1997*; *Hart et al., 1999*; *Tobin et al., 2002*). We examined animals in the presence of high osmolarity using a modified dry drop assay (*Hart et al., 1999*; *Hilliard, Bargmann & Bazzicalupo, 2002*). Wild-type animals responded to encountering the high osmolarity solution by initiating backwards locomotion, while *osm-10* mutants did not respond (*Hart et al., 1999*; *Hilliard, Bargmann & Bazzicalupo, 2002*) (Fig. 5A). *dmd-10* mutants showed a small but significant decrease in their response to high osmolarity as compared to wild-type (Fig. 5A). While *dmd-10*, unlike *osm-10*, is evidently not required to initiate a reversal in response to encountering high osmolarity, the decreased probability of responding in

*dmd-10* mutants suggests that *dmd-10* does play a role in modulating these behavioral responses.

The *dmd-10* mutants exhibit decreased behavioral responses to two different aversive stimuli mediated by ASH: gentle nose touch and high osmolarity. Normal ASH responses to these stimuli require both proper sensation of the aversive inputs and outputs to downstream parts of the neuronal circuit via glutamate signaling. To uncouple these two processes and gain insight into the nature of the behavioral defects in *dmd-10* mutants, we bypassed typical activation of specialized receptors on ASH sensory cilia. Expression of the light-activated ion channel channelrhodopsin 2 specifically on ASH neurons allowed us to use optogenetics to directly activate these neurons and downstream reversal responses with blue light (*Ezcurra et al., 2011*). Notably, mutants with defective sensory cilia that preclude normal nose-touch induced behavioral responses still reliably respond to such optogenetic activation of ASH (*Guo, Hart & Ramanathan, 2009*; *Luth et al., 2021*). The EAT-4 vesicular glutamate transporter is required for glutamate release from ASH, and *eat-4* mutants are defective in glutamate-dependent behaviors such as the nose touch response (*Lee et al., 1999*). We used *eat-4* mutants as a control for the optogenetic stimulation assay and found, as expected, that these mutants were unable to behaviorally respond to optogenetic stimulation (*Luth et al., 2021*) (Fig. 5B). *dmd-10* mutants also showed significantly decreased reversal responses to optogenetic stimulation of ASH as compared to wild type, although not to the same extent as the *eat-4* mutants (Fig. 5B). Since optogenetic stimulation bypasses the need for sensory activation, our data are consistent with a model in which DMD-10 regulates multiple ASH-mediated behavioral responses downstream of sensory stimulation.

## DISCUSSION

We set out to determine whether the transcription factor DMD-10 plays a role in regulating normal neuronal function. In order to address this question we surveyed behaviors in *dmd-10* mutants. We did not observe any obvious changes in locomotion in *dmd-10* mutants, and they exhibited a normal rate of body bends (Fig. 1B) suggesting motor neurons are functioning properly in these mutants. Additionally, we did not find any evidence for defects in the egg-laying circuit (Fig. 1C), the sensory circuits that mediate chemotaxis (Figs. 2A and 2B), or gentle touch to the body (Fig. 2C). The normal responses in these behaviors suggests that DMD-10 is not broadly required for neuronal function.

We did find that three independent *dmd-10* mutant alleles display a decreased responsiveness to nose touch (Fig. 3). Appropriate behavioral responses to stimuli require signaling through complex circuits of neurons. The initiation of a reversal in response to nose touch first requires that the touch be sensed appropriately. Nose touch is sensed, in part, by the pair of ASH sensory neurons. ASH neurons then send signals to several interneurons including the command interneuron AVA. These interneurons signal to DA/VA motor neurons, which synapse onto muscles and initiate backwards locomotion (*Gjorgjieva, Biron & Haspel, 2014*; *Hart, Sims & Kaplan, 1995*; *Maricq et al., 1995*) (Fig. 6). A defect in nose touch response behavior could arise from a defect anywhere in this circuit.

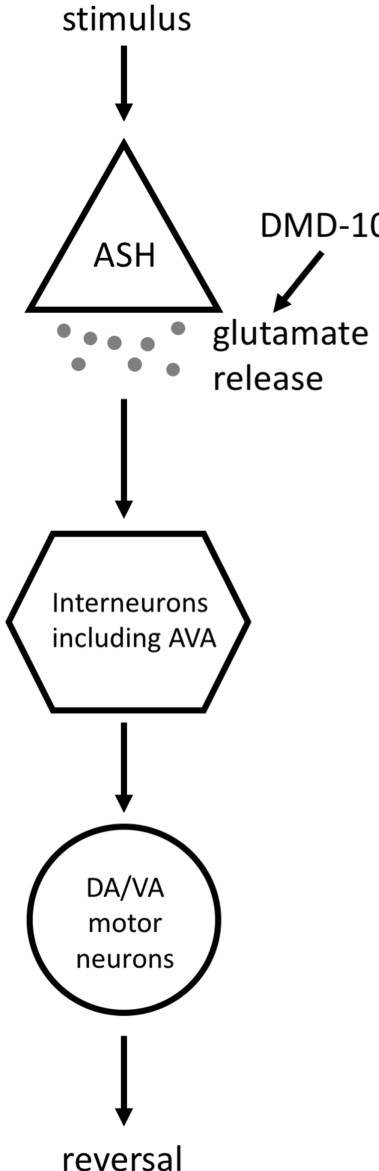

**Figure 6 Model for the DMD-10 site of action.** A simplified model of the signaling pathway that leads to ASH-mediated reversals. Stimuli that are sensed by ASH (in this article we test nose touch and high osmolarity, and use optogenetic stimulation of ASH) cause the release of glutamate from ASH. These glutamatergic signals are received by interneurons, including the command interneuron AVA, which send signals to DA/VA motor neurons that cause the initiation of a reversal. Our data supports a model in which DMD-10 acts to promote glutamate release by ASH.

It is unlikely that a defect in the motor neurons, or in muscles, accounts for the nose touch defect that we see in *dmd-10* mutants. The results from the body bends assay suggest that the motor neurons and muscles are functional (Fig. 1B). Additionally, gentle touch to the anterior of the body elicits normal reversals in *dmd-10* mutants (Fig. 2C). These normal responses to gentle touch to the body also suggest that there is no gross defect in the AVA/AVD command interneurons that are required for reversal initiation since these neurons are required for normal responses to anterior body touch (*Chalfie et al., 1985*).

However, the gentle touch results alone do not rule out an interneuron defect since there are differences in how nose touch and gentle touch to the body activate the command interneurons to initiate backwards locomotion. For example, GLR-1 is required for nose touch, but not gentle touch to the body (*Hart, Sims & Kaplan, 1995*), instead the touch receptor neurons form gap junctions with the command interneurons (*Chalfie et al., 1985*).

Response to nose touch requires expression of the glutamate receptor GLR-1 in interneurons, and a simple explanation for the decreased responsiveness to nose touch in the *dmd-10* mutants would be a decrease in functional GLR-1. Our GLR-1::GFP imaging data shows that there is no change in the quantity of GLR-1::GFP in the ventral nerve cord (Figs. 4A–4D). By itself, this imaging data does not rule out a GLR-1-defect in the *dmd-10* mutants as the cause for the decreased nose touch responsiveness. Processes that affect cell membrane localization of GLR-1 can show behavioral defects, while still showing little or no decrease in the ventral nerve cord GLR-1::GFP levels (*Luth et al., 2021*; *Zhang et al., 2012*). Hypothetically, a nose touch defect in *dmd-10* mutants could still be observed if there was no decrease in the total GLR-1 in the ventral nerve cord, but rather a change in the proportion of GLR-1 that is on the cell surface. We tested this possibility using a strain expressing a GLR-1 variant (GLR-1(A/T)) that is active even in the absence of stimulating input and results in an increase in spontaneous reversal frequency compared to wildtype (*Zheng et al., 1999*, *2004*). This increase is suppressed in mutants with selective reductions in cell surface levels of GLR-1 or with non-functional GLR-1 (*Zheng et al., 2004*; *Luth et al., 2021*). Accordingly, if the *dmd-10* mutation caused a decrease in the population of GLR-1 receptors that are at the cell surface and able to signal we would expect to see a decrease in the rate of reversals in GLR-1(A/T)-expressing animals bearing a *dmd-10* mutation. However, we did not see a decrease in the rate of spontaneous reversals in *dmd-10* mutants expressing GLR-1(A/T) (Fig. 4E), suggesting that in the *dmd-10* mutants the GLR-1(A/T) protein is appropriately localized to the cell membrane, and is able to signal. Thus, taken together our data suggests that the behavioral defect that we observed in the *dmd-10* mutants is unlikely to be caused by altered synaptic GLR-1.

Since we did not find evidence that the decreased nose-touch responsiveness in *dmd-10* mutants was due to changes in GLR-1 receptors in interneurons, we focused our attention on the ASH sensory neurons. ASH neurons are a pair sensory neurons that mediate avoidance of several nociceptive stimuli including touch to the nose, high osmolarity, pH and other aversive chemicals (*Bargmann, Thomas & Horvitz, 1990*; *Hilliard, Bargmann & Bazzicalupo, 2002*; *Hilliard et al., 2004*; *Kaplan & Horvitz, 1993*; *Troemel et al., 1995*). We found that *dmd-10* mutants displayed decreased responsiveness to a second ASH-mediated aversive response—high osmolarity (Fig. 5A). Thus, we see defects in two different ASH-dependent behaviors: mechanosensory response to nose touch and a more modest decrease in chemosensory response to high osmolarity. The detected defects in ASH-dependent behavior are likely not due to decreased sensitivity for inputs as we found that *dmd-10* mutants have decreased responses to direct optogenetic stimulation of ASH (Fig. 5B) that bypasses sensory stimulation. While there are other potential models
that could explain this combination of defects, the most parsimonious explanation is a model in which DMD-10 is a positive regulator of glutamate release by ASH (Fig. 6).

The major way that ASH is believed to encode differences in stimuli is through graded glutamate release and the resultant activation of different sets of glutamate receptors (*Mellem et al., 2002*). Indeed, this difference in glutamate release for different stimuli is consistent with our data. We see a larger behavioral defect for stimuli that elicit modest levels of glutamate release—nose touch (Fig. 3) and the conditions we used for optogenetic stimulation of ASH (Fig. 5)—and a smaller behavioral defect in response to a stimulus that elicits release of higher levels of glutamate—high osmolarity. We speculate that in *dmd-10* mutants, a reduction in glutamate release caused by stimuli that normally evoke greater neurotransmission may still result in enough released glutamate to induce more frequent reversal responses. Interestingly, the different aversive stimuli that are sensed through ASH can elicit different types of reversal responses. Relatively mild stimuli (nose touch, low concentrations of quinine, etc.) elicit short reversals (*Zou et al., 2018*), while harsher stimuli (high osmolarity, high concentrations of quinine, etc.) elicit reversals of longer duration that are coupled with suppression of feeding behaviors (*Piggott et al., 2011*; *Zou et al., 2018*). In this study we have determined whether or not individuals responded to the stimuli tested by reversing, but have not characterized the extent of behavioral responses. It will be interesting to further study the behavioral deficits in *dmd-10* mutants. If *dmd-10* does indeed affect glutamate release we would predict that the character of the response to encountering high osmolarity would be altered in the *dmd-10* mutants such that is more similar to the behavioral response to milder stimuli.

While we can only speculate on the exact mechanism that is affected in the *dmd-10* mutants, their reduced responsiveness to light-induced depolarization yet apparently normal GLR-1 function suggest a defect in presynaptic machinery. There is precedence for this type of mechanism within the doublesex/Mab-3 domain family. *dmd-3* is required for the upregulation of the vesicular glutamate transporter EAT-4 that occurs in PHC neurons in *C. elegans* males (*Serrano-Saiz et al., 2017*). A similar mechanism could operate in ASH where *dmd-10* is required for maintaining the appropriate levels of synaptic proteins. Future work will be needed in order to confirm the site of action for DMD-10 and to investigate the molecular mechanisms that contribute to its behavioral regulation.

## CONCLUSIONS

Our study shows that the transcription factor DMD-10 is important in regulating the responsiveness of *C. elegans* to two different aversive stimuli—touch to the tip of the nose, and high osmolarity. The decreased responsiveness in *dmd-10* mutants is not likely to result from a defect in the localization or function of the glutamate receptor GLR-1. Since both aversive stimuli are detected by the ASH sensory neuron and *dmd-10* mutants show a reduced response to optogenetic activation of ASH, it is possible that DMD-10 is required for appropriate neurotransmitter release by ASH. Future studies will help to elucidate how DMD-10 affects these behavioral responses.

# ACKNOWLEDGEMENTS

We would like to thank the following for strains: Peter Juo, William Schafer, Joshua Kaplan, and the *Caenorhabditis* Genetics Center (CGC). Some strains were generated by the *C. elegans* Gene Knockout Project at the Oklahoma Medical Research Foundation, which was part of the International *C. elegans* Gene Knockout Consortium. We thank Peter Juo and Celeste Peterson for critical reading of the manuscript.

## Funding

This work was supported grants from the CTSD at Suffolk University to Annette M McGehee. Some strains were provided by the CGC, which is funded by the NIH Office of Research Infrastructure Program Grant P40 OD010440. The funders had no role in study design, data collection and analysis, decision to publish, or preparation of the manuscript.

## Grant Disclosures

The following grant information was disclosed by the authors:
Suffolk University.
NIH Office of Research Infrastructure Program: P40 OD010440.

## Competing Interests

The authors declare that they have no competing interests.

## Author Contributions

- Julia Durbeck conceived and designed the experiments, performed the experiments, analyzed the data, prepared figures and/or tables, authored or reviewed drafts of the paper, and approved the final draft.
- Celine Breton performed the experiments, analyzed the data, authored or reviewed drafts of the paper, and approved the final draft.
- Michael Suter performed the experiments, analyzed the data, authored or reviewed drafts of the paper, and approved the final draft.
- Eric S. Luth conceived and designed the experiments, performed the experiments, analyzed the data, prepared figures and/or tables, authored or reviewed drafts of the paper, and approved the final draft.
- Annette M. McGehee conceived and designed the experiments, performed the experiments, analyzed the data, prepared figures and/or tables, authored or reviewed drafts of the paper, and approved the final draft.

## Data Availability

Raw data are available in the Supplemental Files.

## Supplemental Information

Supplemental information for this article can be found online at http://dx.doi.org/10.7717/peerj.10892#supplemental-information.

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
