# Peer review of "The Doublesex/Mab-3 domain transcription factor DMD-10 regulates ASH-dependent behavioral responses"

_PeerJ, doi:10.7717/peerj.10892_

## Round 0.1 · original submission · Major Revisions

Please carefully address the critiques of both reviewers and revise your manuscript accordingly.

Reviewer 1 ·

Basic reporting

Generally well-written report - see General Comments for specific issues. Appropriate citations. Professional overall structure and figures; however, some figures that include dots to represent individual animals (i.e., figure 3) confusing due to number of animals tested - animal dots looks like lines.

Experimental design

This study is a rational design of a behavioral screen. Some details missing from Methods - see General Comments for details.

Validity of the findings

Study provides a foundation in the behavioral effects of dmd-10 mutation; some interpretations move beyond the data (see General Comments for specifics)

Additional comments

Durbeck et al. investigated the possible role of the doublesex/Mab-3 Domain transcription factor DMD-10 in neuron function in C. elegans by performing a behavioral screen using assays for which specific neural circuitry has previously been identified. From these behavioral studies, Durbeck et al. provide evidence that although dmd-10 is expressed in several neurons types, it has a specific role in glutamate release from the ASH neuron. These studies have been carefully designed, though some required changes remain. As a whole, this study could provide insight for future research into the mechanism and location of action of DMD-10.

Major issues:

1) Despite conducting the egg-laying assay on the same plate across genotypes, and therefore animals experience identical testing conditions, a repeated measures (Within Subjects) statistical analysis is not warranted. At best, this allows for a well-controlled experiment, but as each animal does not go undergo each of the genetic conditions being compared a between-subjects analysis is still required. For this study, the statistics need to be reanalyzed using an ANOVA with appropriate post-hoc analyses (similar to the other tests in this paper).
2) For the GLR-1::GFP imaging studies, the researchers do not provide the precise anatomical location from where this construct was imaged in animals. The anatomical location of these images should be described and or illustrated. As well, authors must confirm that each of the images from every animal was captured from this same location across animals. GLR-1::GFP expression can vary greatly along the ventral nerve cord, with very large aggregates occurring more posteriorly and more punctate expression seen more anteriorly. Once the location of the GLR-1::GFP expression has been identified it should also be followed with a citation supporting the selection of that location in the ventral nerve cord.
3) With regards to the nose touch assay as well as the gentle touch assay, the researchers need to provide details about how much time passes between each touch delivery. Also as the authors are reporting an average of 10 touches, it needs to be noted if the worms are as responsive to the first touch as they are to the last touch. If the responsiveness of worms changes over the course of the 10 nose touches, or the 10 gentle touches, then the average responsiveness (% response) for each touch should be reported in a line graph instead of the average of all touches across animals. If the response from first to last touch is consistent then an average would be justified, however, if this is the case it would suggest that the dmd-10 mutant may have a habituation deficit if it shows no decrement in response over 10 touches (see Giles and Rankin, 2009).
4) For figure 3, two of the dmd-10 mutant strains demonstrate a moderate decrease in percent response to 10 nose touches compared to WT while the third strain shows a dramatic decrease in percent response. Is the dmd-10(gk1125) strain responsiveness level significantly lower than the other two dmd-10 strains? The authors should address what could be the cause of these differing results between mutant strains. This would be more interesting to see if those three strains are showing a difference in responsiveness over the 10 touches, rather than speculating from the average across trials.
5) The interpretation that ASH-mediated odor avoidance is affected by the dmd-10 mutation seems somewhat overstated (figure 5a). It's interesting that this slight decline is significant statistically but it's difficult to suggest that DMD-10 and OSM-10 are equal in their impact on ASH neuron function given the dramatic decline in behavior seen with the osm-10 mutation. Further, in figure 5B showing results of optogenetic activation of ASH, the dmd-10 strain appears to be significant both from wild type and from eat-4, so DMD-10 is likely not necessary for glutamate release from ASH. Given that the largest effect for dmd-10 mutation on behavior is seen with the nose touch reversals, it suggests that perhaps the role of DMD-10 may be in fact more pathway-specific within the ASH neuron. As such, the final sentences of the abstract need to be adjusted to reflect this as DMD-10 is not necessary for all ASH-mediated behaviors. As well, this should be included in the discussion section.

Minor Issues:

1) Line 42 - provide references to support “While some DMD family members have been extensively studied,..”
2) Line 55-re-word “We sought to determine whether DMD-10 is required for normal neuronal signaling in…” as neuronal signaling was not directly investigated in this study. Rather, describe the use of a behavioral screen in order to denote neuron function.
3) Line 60 - provide a reference to support sentence “Behaviors such as egg-laying, chemotaxis, and body bending are associated with signaling at particular synapses.”
4) Line 82-correct the sentence with “encode difference and stimuli”
5) Line 110-give the diameter of the copper ring employed in the egg-laying assay
6) Line 118 - replace with correct statistical analysis, within-subjects design (repeated measures) is not justified for this study (see Major issue #1 above)
7) Nose touch section (lines 130-138) - describe the amount of time that occurs between touches, is it a consistent time period between touches, and what determined the amount of time between touches? Also, report that it was the average across 10 touches that was reported. Ideally, report responsiveness for each individual touch to demonstrate if a plasticity process is at play. (see Major Issue #3 above for more on this)
8) Gentle touch section (lines 141 - 146) – the same as the nose touch section, describe how much time passed between gentle touches and if the averages were reported.
9) For the dry drop assay (lines 149-155) - describe how much time occurs between each dry drop encounter and if the responsiveness changes over time (account for possible adaptation).
10) For Optogenetic Activation of ASH (lines 158 - 168) - provide justification that testing off ATR did does not impact the function of the channelrhodopsin (reference needed). Also, provide measured light intensity that results from the LED setup described. Again, report duration between each of the 10 light stimuli delivered and responsiveness across 10 stimuli. Is there sufficient time between each light presentation to ensure transmitter rundown does not occur? Provide citation to support.
11) Line 177-178, correct wording of sentences “worms that had been laid 3 days prior to the experiment. Worms 178 were twice with M9 buffer and once with water.”
12) Line 178 - Include water type used for the third wash. If this is a typical assay procedure, provide citation.
13) Line 228 begin new paragraph after “...these mutants.”
14) Line 322 – re-word statement “These data suggest that the defect in dmd-10 mutants is not due to a decrease in the sensation of nociceptive stimuli by ASH.” - this was not demonstrated as cell activity of ASH was not measured.
15) Figure 1 caption - Include corrected statistical analysis description.
16) Figure 5A, the moderate decrease in responsiveness to the dry drop assay of the dmd-10 mutant is likely statistically significant from osm-10 outcome (see Major issue #5 above).
For figure 5B, is eat-4 significant from both wild type and dmd-10 mutants? This should be reported and these moderate results discussed.

References:
Giles, A. C., & Rankin, C. H. (2009). Behavioral and genetic characterization of habituation using Caenorhabditis elegans. Neurobiology of learning and memory, 92(2), 139-146.con

Reviewer 2 ·

Basic reporting

The authors provided sufficient background and the goals of the study were made clear. The data provided including the figures and raw data was well organized and clear. The amount of background or rationale as it relates to the experiment or assay needs to be expanded in certain places, but is mainly clear. I have highlighted where it is not. Please see general comments section.

Experimental design

In their manuscript “The Doublesex/Mab-3 domain transcription factor DMD-10 regulates ASH-dependent behavioral responses “the authors did a thorough investigation of the role of dmd-10 neuronal function in C. elegans using a combination of behavioral assays and optogenetic techniques. The goals of the study align with the scope of the journal and the research proposed and completed fill a gap in the field.

Validity of the findings

The authors determine that dmd-10 gene function is required for ASH mediated behaviors and the experiments assessing GLR-1 expression at the cell surface versus the ventral nerve cord in dmd-10 mutants and wild type animals adds a new level of detail of the role of dmd-10 in neuronal function. The authors completed many behavioral assays with sufficient replicates with statistical tests to definitely determine the role of the dmd-10 in neuronal function. The raw data provided is well organized and easy to understand and could be replicated.

Additional comments

In their manuscript “The Doublesex/Mab-3 domain transcription factor DMD-10 regulates ASH-dependent behavioral responses “the authors did an investigation of the role of dmd-10 neuronal function in C. elegans using a combination of behavioral and optogenetic techniques. The authors determine that dmd-10 gene function is required for ASH mediated behaviors and the experiments assessing GLR-1 expression at the cell surface versus the ventral nerve cord in dmd-10 mutants and wild type animals adds a new level of detail of the role of dmd-10 in neuronal function.

Overall, the study is well designed, experiments and analysis complete. The findings are novel and clearly fills a gap in knowledge and establishes a role for dmd-10 in regulating ASH behaviors.

The manuscript is mostly clear, but there are a few areas where the rationale could be better explained or additional information provided to make better connections. I expand and provide examples below. I also note a few places where there are minor errors or provide general feedback. Please see my comments below.

Avoid using worm, either use nematode, animal, or C. elegans.

Lines 56-67, the authors mention that they are using behavioral studies to determine where in the nervous system DMD-10 functions. The authors need to provide a better rationale for how using the behavioral assays helps determine where DMD-10 function is required. In other words, make a connection with the genes in the behavioral circuit are in a pathway and that you are using the behavioral assay to determine where loss of dmd-10 functions.

The authors bring up ASH neuron specific behaviors but the transition is a little rough. And the level of background information provided for ASH and regulation of different responses in behavior is extensive and distracting. I suggest simplifying. For example, lines 77-81 the information provided does not further the authors rationale.

Lines 212-213: “We are interested in understanding the role of the Doublesex/Mab domain transcription factor DMD-10.” In what? A role for neuronal development, neuronal function as it relates to behavior? Be specific

The author needs a better introduction to each of the behavioral assays and why authors are testing certain other mutants associated with the assays for controls. For example, the trashing assay is used to test for defects in motor neuron control. Then the author goes on to mention egl-3 and the egl-3 mutant defects. The author needs to mention how egl-3 is involved with the circuit. Why did you test this mutant? This is the case for all the behavioral assays.

Lines 226-227, the authors need to state what is being compared. For example, no difference compared to wild type.

Lines 228-229, “Along with unaltered behavior driven by motor neurons controlling locomotion, we also found no defect in the motor control of egg laying.” I am not sure what the authors mean by motor control of egg laying. This could be better stated, by saying the neural circuit responsible for egg laying was not defective. Again, this needs a little more detail. How does this behavioral experiment differ from motor neuron/thrashing experiments? A few lines on the neural circuit for egg laying would help clear things up.

Lines 282-287, the authors make the argument that there was no change in the GLR-1::GFP at the synapses or a better explanation for how the constitutively active GLR variant helps discriminate between altered cell surface GLR-1 and presynaptic GLR-1.

Line 287, functional and properly-localized GLR-1: You are missing a word here or need to remove the semicolon.

In lines 315-320 the authors discuss EAT-4 mutant, but do not mention the phenotype expected. Make a connection between nose touch and reversal response.

In general, for the discussion, discuss the results in order. The authors go from discussing working model in figure 6 and discussion of the results to figure 2C. For example, in lines 333-335, the authors discuss figure 2C, before figures 2A-B.


Lines 358-373, In the discussion the writing around total quantification of GLR-1at the ventral cord not altered in dmd-10 mutants and the potential hypotheses for how loss of dmd-10 could be affecting ASH dependent behaviors, is not clear. I would suggest stating what you found, then present the possible scenarios, followed by how you tested to gain insight into what was happening, and then what you found. I would then explain what this suggests.

Line 366, there is an extra () at the end of the sentence.

---

## Round 0.2 · Minor Revisions

Please address remaining concerns of the reviewers and amend your manuscript accordingly.

Reviewer 1 ·

Basic reporting

The reporting has improved from the previous submission in that the data analysis has been corrected and data display has been improved. There is greater clarity in the figure captions of the result reported in the figures and raw data has been included. Overall writing-style is clear and information has been added to help explain the study approach and the rationale for results.

Experimental design

Experimental design of a behavioral screen followed by a specific receptor expression assay and a neuron activation study are sound approaches to the questions of the study. There is more research to be performed in the future to support the proposed rationale in the discussion.

Validity of the findings

Findings appear valid; however, inclusion of the third dmd-10 mutant strain previously reported is recommended given that the data has been collected, and this appears to be a venue that is amenable to inconclusive results. A rationale should be provided for the different outcomes seen with the third strain though.

As well, how/where strains were procured needs to be provided in the Materials and Methods section. As the dmd-10 mutant strains were originally generated by the Knockout Consortium, this needs to be recognized as per the requirements of using these strains. As well, details of the number of backcrosses performed on each of these strains need to be included.

Additional comments

Overall, the results are clear that there is an effect on behavior in the dmd-10 mutant strains. For clarity, it is necessary to distinguish the statements regarding dmd-10 effects on glutamate release in the Discussion as being speculative as this was not directly tested. Specific minor comments below:

Line 475 - "The detected defects in ASH-dependent glutamate signaling..." - reword as ASH-dependent glutamate signaling was not directly tested in this study.

FIgure 2C. Missing confidence interval for dmd-10

Reviewer 2 ·

Basic reporting

Overall, this is a much stronger manuscript and most of the comments, suggestions, and issues that I raised in the original submission were addressed in this resubmission.

The background section was greatly improved with a better introduction of dmd-10 and the overall approach to understand the role of dmd-10 in regulating C. elegans behavior.

Experimental design

The rationale for the experiments was improved by the addition of supporting details and detailed explanation of the experimental design

Validity of the findings

The data and figures provided are statistically sound and well controlled. The figures have been modified and improved. Conclusions are well supported.

---

## Round 0.3 · accepted · Accept

Thank you for addressing the remaining critiques of the reviewer. I am please to accept revised manuscript now.